# Characteristics of Distinct Dietary Patterns in Rural Bangladesh: Nutrient Adequacy and Vulnerability to Shocks

**DOI:** 10.3390/nu13062049

**Published:** 2021-06-15

**Authors:** Zakari Ali, Pauline F. D. Scheelbeek, Kazi Istiaque Sanin, Timothy S. Thomas, Tahmeed Ahmed, Andrew M. Prentice, Rosemary Green

**Affiliations:** 1Nutrition Theme, MRC Unit the Gambia at the London School of Hygiene and Tropical Medicine, Atlantic Boulevard, Fajara, P.O. Box 273, Banjul, The Gambia; Andrew.Prentice@lshtm.ac.uk; 2Faculty of Epidemiology and Population Health, London School of Hygiene and Tropical Medicine, Keppel Street, London WC1E 7HT, UK; Pauline.Scheelbeek@lshtm.ac.uk; 3Centre on Climate Change and Planetary Health, London School of Hygiene and Tropical Medicine, Keppel Street, London WC1E 7HT, UK; 4Nutrition and Clinical Services Division, International Centre for Diarrhoeal Disease Research, Dhaka 1212, Bangladesh; sanin@icddrb.org (K.I.S.); tahmeed@icddrb.org (T.A.); 5International Food Policy Research Institute, 1201 Eye Street, Washington, DC 20005, USA; tim.thomas@cgiar.org

**Keywords:** staples, dietary pattern, latent class analysis, farm production, diet vulnerability, nutrition transition

## Abstract

Food security in Bangladesh has improved in recent years, but the country is now facing a double burden of malnutrition while also being highly vulnerable to climate change. Little is known about how this may affect food supply to different sectors of the population. To inform this, we used a national dietary survey of 800 rural households to define dietary patterns using latent class analysis. Nutrient adequacy of dietary patterns and their potential vulnerability to climate shocks (based on diversity of calorie sources) were assessed. We fitted mixed effects logistic regression models to identify factors associated with dietary patterns. Four dietary patterns were identified: rice and low diversity; wheat and high diversity; pulses and vegetables; meat and fish. The wheat and high diversity and meat and fish patterns tended to be consumed by households with higher levels of wealth and education, while the rice and low diversity pattern was consumed by households with lower levels of wealth and education. The pulses and vegetables pattern was consumed by households of intermediate socio-economic status. While energy intake was high, fat and protein intake were suboptimal for all patterns except for the wheat and high diversity pattern. All patterns had fruit and vegetable intake below the WHO recommendation. The wheat and high diversity pattern was least vulnerable to shocks, while the rice and low diversity pattern was the most vulnerable, relying mainly on single cereal staples. The diets showed “double vulnerability” where the nutrient inadequate patterns were also those most vulnerable to shocks.

## 1. Introduction

Bangladesh is one of the most populous countries in the world and will be home to a significant proportion of the world’s population in the near future despite declines in population growth rates over the last decade [1]. The country has undergone one of the most rapid declines in poverty in the world in the last 30 years, and this has been accompanied by major improvements in food security. However, this progress has not occurred equally across all population groups, and large sections of the population are still vulnerable to food insecurity [2]. The country is, thus, currently facing a double burden of malnutrition where both over- and under-nutrition persist in the same society [3,4]. Recent estimates show that 31% of children under five years are stunted [5], yet overweight and obesity prevalence is above 18% in adults and has tripled in adolescents between the years 2000 and 2016 [6].

Given the diversity of diets and nutritional health within the country, the requirements and needs for nutritional improvement vary greatly [7]. Thirteen per cent of the population have insufficient habitual food consumption, and 32% experience moderate or severe food insecurity [8]. In addition, Bangladesh is classified as being highly vulnerable to climate change [9] and has widespread subsistence agriculture [10], which adds further pressure to food production, supply and food prices. Due to the substantial inequalities in access to an adequate and diverse supply of foods, some diets may be more vulnerable than others to shocks such as price fluctuations induced by climate change, disease outbreaks or other events.

In order to take evidence-based decisions aiming at improving nutritional health and resilience to shocks at the sub-national level, it is important to identify hotspots of poor nutrition (both over and under-nutrition) and study the exact composition and socio-economic determinants of such diets. Identification of dietary patterns allows for a broader understanding of population diets than looking at individual foods or nutrients alone [11,12]. Analysis of single nutrients or foods also fails to account for the fact that people eat meals that include many foods and nutrients that interact in different ways. This makes dietary advice based on these analyses less useful and more difficult to implement. As a result, population-wide dietary pattern analysis has been proposed as an alternative method of assessing the way people eat and its relationship with health [13]. The identification of dietary patterns over space and time can be a solid first step in identifying what needs there are at sub-national levels, so as to guide key decision makers to develop tailored responses. Patterns can identify which food groups are inadequate or overly abundant in diets and identify priority populations at risk of poor nutrition. This can inform the planning of agriculture and food production to improve diets.

In this analysis, we have determined dietary patterns, identified associated factors and assessed both nutrient adequacy and vulnerability of the diet to potential shocks (based on the diversity of calorie sources) in order to inform appropriate diet shifts towards resilient and nutrient-adequate diets in Bangladesh.

## 2. Materials and Methods

### 2.1. Study Design and Participants

This study is a secondary analysis of cross-sectional data from the Bangladesh Climate Change Adaptation Survey (BCCAS) Round I [14]. The survey involved 800 farming households, which were selected to be representative of the seven broad agroecological zones of Bangladesh and collected in the same season. It was administered by the International Food Policy Research Institute (IFPRI) and Data Analysis and Technical Assistance Limited (DATA) from December 2010 to February 2011. The database contains data on demographic characteristics, crop and livestock production, food consumption, land tenure, incidence of climatic shocks in the last five years and adaptation options. The present analysis is based on the food consumption, crop and livestock production and household characteristics data.

### 2.2. Food Consumption and Socio-Demographic Data

Participants reported through interview, the quantities of household consumption of 127 different food items over a 14-day recall period. Women responsible for food preparation and distribution were interviewed on behalf of household members with assistance from other household members. They were asked to estimate the amounts in grams or kilograms of foods household members had consumed over the last 14 days preceding the survey. Total household level intakes of each raw food item in grams were linked with the Bangladesh food composition tables (FCT) [15] to determine total energy and amounts of carbohydrate, fat and protein. Subsequent calculations of energy from carbohydrate, fat and protein were done by multiplying total amounts in grams of each macronutrient by the general Atwater factors: 4, 9 and 4 kcal/g, respectively.

We obtained proxy-individual level daily intakes of energy and macronutrients by dividing daily household intakes by a weighted sum of the household size, which accounted for age and sex differences in energy intake. We used proxy-individual requirements that are relevant to South Asia and previously used by the Indian National Sample Survey [16] and recently by Aleksandrowicz et al. [17] to convert daily household level intakes into approximate individual intakes. This approach has been demonstrated to be a useful measure, which approximates well the individual intakes from household level data [18].

The adequacy of energy intake from macronutrients was calculated following World Health Organization (WHO) guidelines on the appropriate daily range of proportions of energy from macronutrients to total energy intake [19]. We calculated the proportion of energy each macronutrient contributed to the total daily energy intake and compared it with the WHO guideline. Participants whose proportion of intake of a macronutrient fell within the WHO recommended range for that nutrient were regarded as having adequate energy intake for that nutrient.

Other covariates selected based on previous literature were household size and age, educational level and religion of the head of household. Variables representing characteristics of the head of household were chosen on the basis that they would act as a proxy for general household characteristics. Household wealth status was determined using household possession of 29 durable items in principal component analysis [20,21].

Farm or home production diversity was assessed as the total number of types of crops and livestock produced by households over the last 12 months preceding the survey [22]. We classified farm production diversity into ≤5, 6–10 and >10. Grouping of categorical variables was done to ensure a reasonable spread of participants in each category in order to avoid the problem of data sparsity in statistical modelling.

### 2.3. Vulnerability of Dietary Patterns

The vulnerability of household dietary patterns was assessed based on extent of dependency on major calorie sources of the diets [23]—an indication of dietary vulnerability to crop failure or price fluctuations. Vulnerability scores were calculated by determining calorie concentration proportions of staples to total consumption in the diets. Dietary patterns with the highest proportion of calories from single staples were considered to be more vulnerable than those dependent on many major sources of energy.

### 2.4. Statistical Analysis

Thirty-two food groups were created from the 127 food items based on shared nutritional content. For example, different types of large fish and small fish were combined to form the large and small fish food groups; similarly, different types of green leafy vegetables were combined to form the green leafy vegetables group. Details on the food groupings are provided in the Appendix A (Appendix A). Average household level energy intakes were categorized into zero, low, medium and high proportions of total calories by each food group (i.e., zero intake and tertiles of energy from that food group as a proportion of total dietary energy consumption). The categorical variables were used to define separate dietary patterns of the data using finite mixture modelling. All dietary predictor variables were considered to be uncorrelated with one another and were entered into the latent class analysis (LCA) models (see Appendix A). The LCA method of dietary pattern analysis allows membership of dietary patterns to be mutually exclusive, hence each household was assigned to a single dietary pattern based on probability. We described each pattern by looking at the mean consumption of the different food groups by households assigned to that pattern.

The main explanatory variables were wealth status, age, religion, marital status and educational level of the head of household, plus household size and farm production diversity of households. Dietary patterns were the outcomes of interest.

Due to likely clustering in the data within both households and districts, we fitted mixed effects logistic regression models to enable us to account for the correlations in dietary intake. Therefore, we specified households nested in districts as random effects and included the explanatory variables as fixed effects utilizing the *melogit* command in Stata. Further details on the modelling strategy, tests for interactions and multicollinearity are provided in the Appendix A (Appendix A).

LCA was conducted in Mplus (version 7.5). Descriptive statistics and mixed effects logistic regression modelling were performed using Stata (version 16.1).

## 3. Results

### 3.1. Dietary Patterns in Bangladesh

The study sample included 800 households in predominantly rural areas of Bangladesh. The mean age of household heads was 45 ± 14 years who mostly practiced Islam (88.9%), and nearly half (47.6%) had no formal education. The average household size was 5 ± 2 members, and 40% were classified as poor households. Most households across dietary patterns had farm production diversity between six and 10 different crops or livestock (48.5%). All geographical regions of Bangladesh were reasonably equally represented (Table 1).

Four dietary patterns were identified in the best-fitting latent class analysis model. We named these patterns rice and low diversity (263 households or 33% of the sample); wheat and high diversity (262 households or 33% of the sample); pulses and vegetables (259 households or 32% of the sample); meat and fish (16 households or 2% of the sample), based on the major food groups consumed in each pattern.

### 3.2. Energy and Nutrient Intakes of Dietary Patterns

There were notable differences in the intake of staple cereals between dietary patterns: all patterns were highly based on rice, but rice consumption was highest among participants who followed the rice and low diversity and the meat and fish pattern, while those who followed the wheat and high diversity pattern consumed around 500 kcal less rice per day, accompanied by a higher consumption of wheat, pulses and oils (Figure 1).

Fruit and vegetable consumption was low in the rice and low diversity pattern at around 100 kcal per day and highest in the wheat and high diversity pattern at around 180 kcal per day. Meat and fish represented a significant amount of daily energy for followers of the meat and fish and the wheat and high diversity patterns but were much more rarely consumed in the other two patterns (Figure 1). Dairy products were hardly ever consumed in all sampled households (Appendix A). Further details of food groups that describe the patterns are provided in Appendix A.

The average daily energy intake was 2934 kcal/capita/day. Energy intakes were highest among followers of the wheat and high diversity pattern (3055 kcal/capita/day) and lowest in the rice and low diversity pattern (2813 kcal/capita/day) (Table 2). Average diets across the sample were high in carbohydrates and did not meet WHO recommendations for the proportion of energy from fat or protein (Table 2). However, consumers of the wheat and high diversity pattern did meet these requirements on average, and consumers of the meat and fish pattern just met the recommendation on protein intake. In the rice and low diversity pattern, only 3% of individuals met the requirement on fat intake, and 10% met the requirement on protein intake. Even though participants in the wheat and high diversity pattern had the highest daily fruit and vegetable intake (381 g), this was still below the WHO recommendation of 400 g per day. Only 18.4% of the total sample met the recommendation for fruit and vegetables—lowest in the rice and low diversity pattern (3.4%) and highest in the wheat and high diversity pattern (37.0%) (Table 2). The distribution of proportions of participants meeting the WHO recommendations for carbohydrates, fat, protein and fruit and vegetables by other household characteristics are presented in the Appendix A (Appendix A).

### 3.3. Predictors of Dietary Patterns in Bangladesh

Wealth and education levels both varied according to the dietary patterns: the rice and low diversity pattern tended to be consumed by poorer households and those with lower levels of education, while the meat and fish and wheat and high diversity patterns were consumed by households with higher levels of education and wealth (Table 1 and Appendix A). The pulses and vegetables pattern tended to be consumed by households of intermediate status. There was also some evidence of differences in dietary patterns according to region. In particular, the rice and low diversity pattern was more likely to be consumed in the north and less in the central and southern regions, as was the meat and fish pattern, although the sample size for this diet was low. The wheat and high diversity pattern was less likely to be consumed in the north and more common in the southern and central regions. Few other differences between household characteristics and dietary patterns were observed (Table 1).

In a mixed effects regression model adjusting for other potential predictor variables including household wealth, age of household head, religion, education, household size, farm production diversity and region of residence (Table 3), the most important predictors of consumption of the rice and low diversity pattern were lower household wealth (OR 0.41 (95% CI: 0.27–0.61) of following this pattern for the wealthiest households compared to the least wealthy), lower educational attainment (OR 0.37 (95% CI: 0.18–0.77) for heads of households with at least secondary education compared to none), larger household size (OR 4.15 (95% CI: 2.28–7.58 for largest households compared to smallest) and residence in the northern region (OR 2.43 (95% CI: 1.46–4.03) compared to those in the south) (Table 3).

By contrast, higher household wealth (OR 2.66 (95% CI: 1.77–4.02) for the wealthiest households compared to the least wealthy), smaller household size (OR 0.28 (95% CI: 0.15–0.54) for the largest households compared to the smallest) and residence in the southern region (OR 0.43 (95% CI: 0.26–0.70) for households in the north compared to the south) were the strongest predictors of the wheat and high diversity pattern.

There was no significant association between any household characteristics and households following the pulses and vegetables dietary pattern, which appeared to be broadly spread across all categories of households.

For predictors of the meat and fish dietary pattern, due to data sparsity problems in multivariable models, we present only univariate logistic regression tests of associations between explanatory variables and this dietary pattern. Non-Muslim-headed households were more likely to have this pattern of eating compared with Muslim-headed households (OR 3.79 (95% CI 1.28–11.17)), and having at least secondary education showed some evidence of an association (OR 1.28 (95% CI: 1.65–23.95)). However, results for this pattern should be interpreted with caution due to the small sample size.

In general, age of household head tended to be only weakly associated with dietary pattern, and we found no significant associations between levels of farm production diversity and dietary pattern.

### 3.4. Vulnerability of Dietary Patterns to Potential Shocks

The test of vulnerability of the diets to possible price fluctuations and crop failure showed that diets in the sample tended to be highly vulnerable with an overall 75% of total household calories contributed by a single staple crop (rice). The rice and low diversity pattern was the most vulnerable with 84.2% of total calories contributed by a single staple crop; the wheat and high diversity pattern was the least vulnerable, with a little above half of total calories (63%) derived from a single staple. The pulses and vegetables and meat and fish patterns were the second and third most vulnerable diets, respectively (Table 4).

## 4. Discussion

### 4.1. Summary of Main Results

Our analysis of dietary patterns among households in Bangladesh identified four distinct dietary patterns: we characterized these as being based on rice and low diversity, wheat and high diversity, pulses and vegetables and meat and fish. The most important factors predicting dietary pattern of households were wealth and education, with family size, age and religion also showing some associations. Farm production diversity was not associated with dietary pattern after adjustment for other factors. All patterns attained by far the majority of their energy supply from carbohydrates and tended to have inadequate consumption of the other macronutrients and fruit and vegetables. Households following the rice and low diversity pattern had the lowest proportion of members with adequate fat, protein and fruit and vegetable intake. Those following other patterns had a higher proportion of members with adequate intakes. The rice and low diversity and meat and fish patterns were the most vulnerable diets to external shocks, while the wheat and high diversity and pulses and vegetables patterns were less vulnerable to potential shocks and potentially more nutritionally adequate, as indicated by higher intakes of fruit and vegetables.

### 4.2. Comparison with Previous Studies

Previous studies reporting dietary patterns in Bangladesh have found varying numbers of patterns. Two studies found three patterns using principal component analysis [24,25]. Seven patterns were identified by Waid et al. using principal component analysis [26]. The basic components of identified patterns are, however, similar and include intake of foods such as rice, vegetables, pulses, meats and oils, which are similar to the foods that make up the patterns identified in the present analysis. For instance, the three patterns described by Chen et al. [24] were a “balanced pattern” characterized by rice, fish, meat, fruits and vegetable consumption; an “animal protein diet” and a “gourd and root vegetable diet”. These patterns could be traced to the wheat and high diversity, meat and fish and the pulses and vegetables patterns identified in the present study.

The high wealth and education associated with the wheat and high diversity and the meat and fish patterns make them characteristic of diets that might represent a nutrition transition. Even though these patterns were generally associated with high energy from fats and proteins, the wheat and high diversity pattern may be a healthier pattern than the meat and fish pattern because it had higher intakes of fruit and vegetables. Previous studies have found evidence of a positive association between production diversity and dietary diversity [22,27]. This was not the case in the present study, although for the two patterns more associated with higher social status, there was some evidence of a positive trend with increased production diversity. However, in this study, at least it appears clear that wealth and education levels were much more clearly associated with dietary diversity than diversity of production. There is evidence that as wealth increases, consumption of less expensive staples such as rice decreases with a shift towards intake of prestigious cereals such as polished rice or processed wheat, meat, oils, sugar and high diet diversity [28,29]. The characteristic high education and wealth of these two patterns is indicative of this idea. These findings are consistent with a recent global analysis, which showed that household wealth and higher female education are strongly related to nutrition and health outcomes [30].

Low membership of the meat and fish pattern may suggest that it is an emerging dietary pattern in the country. It has characteristics typical of a nutrition transition diet: high meat, oils, rice and sugar [31]. In addition, low membership could be because the sample was drawn from predominantly rural farming areas of Bangladesh, and most of this pattern’s components are less likely to be available to or affordable by many rural households.

The high proportion of total energy from carbohydrates in all patterns is consistent with the results of previous national data, where 70% of total energy of Bangladeshi diets was from cereals [32]. Even though increased energy intake from carbohydrates has been linked with good health and delayed onset of chronic diseases [33], having more than the recommendation could mean a substitution for other essential energy sources that are important to health. The low proportion of households with adequate intakes of fats, proteins and fruit and vegetables is indicative of nutrient substitution.

### 4.3. Strengths and Limitations

This study has some important strengths. Data were collected on a wide range of household variables across population groups in Bangladesh, which allowed for a better understanding of diet and associated characteristics. Another strength is the availability of food quantities, which enabled energy calculations for use in LCA, improving the interpretations of results compared to using food frequencies to define dietary patterns [34]. An additional strength is the ability to estimate macro-nutrient adequacy of the diets, which many studies reporting dietary patterns are not able to do, either due to limitations in statistical methods or underlying data limitations. Our assessment of the relative vulnerability of the diets to external shocks is an added strength not usually assessed in similar studies.

The study also has some limitations that are worth noting. We cannot rule out data validity and reliability problems associated with use of recalled dietary intake data. However, the longer recall period could mean that the dietary data reflect the usual intake of households, hence dietary patterns resulting from them are more likely to reflect people’s regular food intake. The United Nation’s Food and Agriculture Organization food balance sheet data show an increasing trend of total energy per person per day in Bangladesh. However, these have remained ~2500 kcal/person/day between 2008 and 2013 [35], which could indicate some level of over-reporting of energy intakes in this study where average intakes were ~3000 kcal per person per day. As the estimation of dietary patterns relies largely on comparing proportions of different foods consumed rather than absolute amounts, over-reporting is likely to have had a minimal effect on the findings of the study. Despite the likely high over-reporting of intakes, fats, proteins and fruit and vegetables consumption were still suboptimal, implying that actual intakes could even be lower than what we have reported. There is also a possibility that the results obtained from the survey were seasonally biased, since data collection took place only in the months of December to February. Other dietary patterns may, therefore, exist among this population in other seasons that were not captured in the present analysis.

In addition, the data used in defining dietary patterns did not include foods eaten out of home. It is possible the foods people eat out of home are different from what they will eat in the home. This may constitute an important part of the pattern that people eat and could affect the validity of our results. While this may be a limitation of the data, it can be difficult to obtain accurate estimates of out of home food intake at the household level, and attempts to do this may increase errors associated with recalled data. There are also not many reasons to suggest that out of home eating had much effect on our results, because households were from mostly rural areas and foods eaten away from home are less likely to be very different from what they will normally eat at home.

Further, individual-level energy and macro-nutrient intakes were derived from household level estimates using proxy nutrient requirements of individuals. Even though this approach is shown to closely approximate individual intakes in comparison with self-reported data [18] and has also permitted the use of household food expenditure and food consumption data in making useful individual level inference, some limitations may persist. For instance, household food may not be equitably distributed among all members [36], but the approach is unable to account for this potential intra-household food distribution problem. Therefore, our estimates of energy intake and macro-nutrient adequacies need to be interpreted with caution. To avoid over-interpretation of results, we did not estimate micro-nutrient adequacy of the diets; rather, we assessed the adequacy of fruit and vegetable intake, a good proxy for micro-nutrient intake. However, because of the limited nutritional variables used in the analysis, we were unable to adequately measure over-consumption of particular nutrients such as sugar or salt, which may have shed further light on the double burden of malnutrition in Bangladesh.

When defining the vulnerability of dietary patterns to shocks, we used the proportion of energy that came from single staples as the defining factor. This measure is, therefore, based heavily on diversity of the diet and does not take into account potential vulnerability of individual crops to particular shocks. Some dietary patterns may, therefore, be more vulnerable in particular ways that are not documented in this analysis.

Another limitation is the size and representativeness of the dataset. The data are representative across regions, but there was likely under representation of urban and some socio-economic groups because households were sampled from largely rural farming areas. The sample size proved adequate for the analysis of three of the four dietary patterns identified, but there were too few households assigned to the meat and fish pattern to produce a robust analysis. Finally, the data were collected between 2010 and 2011, so diets may have changed quite substantially since this time.

### 4.4. Implications Including Policy Recommendations

Our findings suggest that energy deprivation is not a major problem among rural farming households in Bangladesh. Public health nutrition efforts should focus on improving other diet quality indicators: fruit and vegetable intake, fats and protein adequacy.

We also find that household wealth and education were more strongly related to diets than farm production diversity. Those who are doing the farm production might prefer selling their products over consumption, as they perceive it as an income generating source. Public health nutrition interventions aiming to improve diets should target the consumption of one’s own produce among less wealthy and less educated households.

Further, the dietary patterns showed characteristics of an emerging nutrition transition. The patterns represent a mixture of both healthy and potentially less healthy diets. They appear to indicate a transition from a more staple-based diet composed of less diversity through a mixture of staples and high diversity towards diets high in meats, fats and refined foods (potentially represented by the emerging meat and fish pattern, although this diet was only consumed by small numbers of households at the time of data collection). These results are suggestive of an emerging transition of Bangladeshi diets towards potentially less healthy patterns as people get wealthier [37], although in our analyses, we found no evidence that fat intake was above the recommended levels in any of the dietary patterns.

Finally, our data show evidence of a double vulnerability of diets [38], whereby the dietary patterns most associated with nutritional inadequacy (rice and low diversity and pulses and vegetables) were also those most vulnerable to shocks including those that may occur as a result of food price fluctuations and future climate change. As Bangladesh is already highly vulnerable to climate change, crop failure and food price instability, especially in rice, it will have a greater impact on diets. In the spectrum of the transitioning diets, the pulses and vegetables diet could be an achievable short-term goal towards a lower vulnerability and increased nutritional adequacy for households currently consuming the rice and low diversity diet, due to its greater diversity and likely higher micronutrient content. While diet diversification of major staples will make diets less vulnerable to shocks from crop failure and price instability, switching to other staples such as wheat or maize could come with additional costs in the short term, which can hinder their uptake and sustainability. In spite of this possible initial setback, our analysis shows that some households such as those following the wheat and high diversity pattern are already deriving a substantial amount of dietary energy from alternative sources such as wheat, which can be considered for a national scale-up. While existing Government of Bangladesh price stabilization plans for rice remain a major guard against diet vulnerabilities, potential future declines in rice yields due to climate change and resulting increases in price may become unbearable in the long term. Therefore, it is important to promote and subsidize alternative staples such as wheat and maize to ensure that major shocks affecting rice will have minimal impacts on overall diets in Bangladesh.

## 5. Conclusions

Our findings show clear evidence of vulnerability to shocks among all diets in the sampled rural Bangladeshi households, due to their reliance on single staple foods. The identified dietary patterns consist of a mixture of healthy and potentially less healthy patterns, with some patterns showing evidence of greater dietary diversity but only among wealthier population groups. The diets also showed “double vulnerability” where the nutrient-inadequate patterns were also those most vulnerable to potential shocks. National food policies should aim to increase the availability of other staples and fruits and vegetables to reduce over reliance on single staples such as rice to minimize dietary vulnerability and improve nutrient adequacy of diets.

## Figures and Tables

**Figure 1 nutrients-13-02049-f001:**
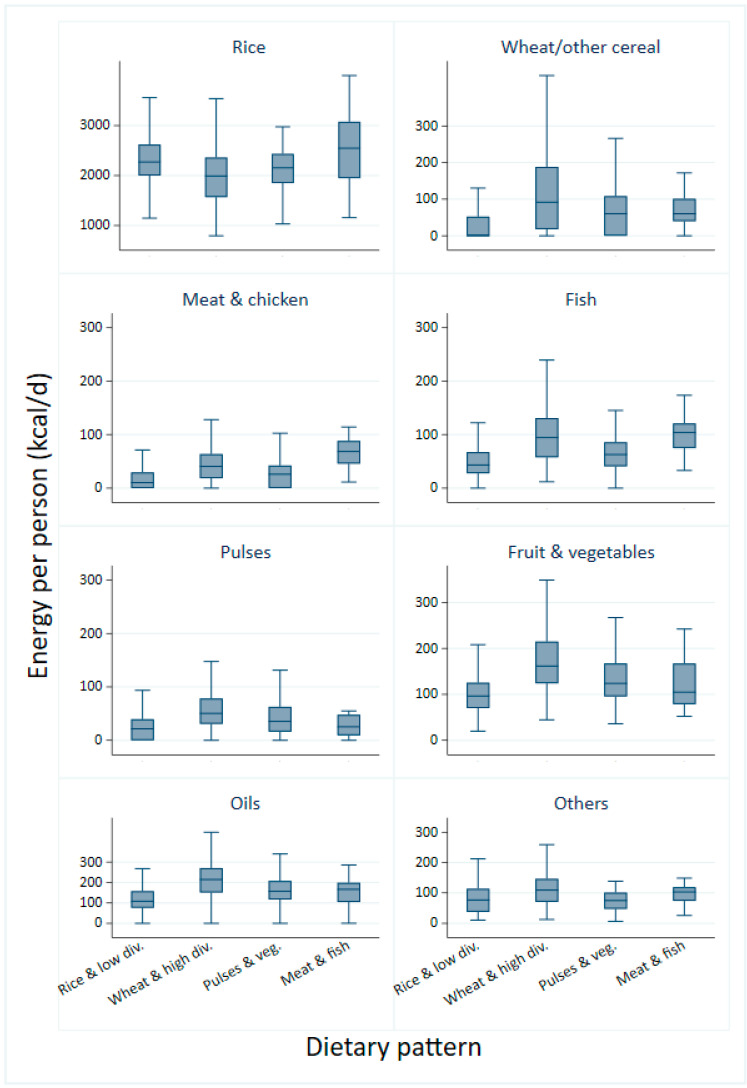
Mean energy from consumption of food groups by dietary pattern.

**Table 1 nutrients-13-02049-t001:** Distribution of socio-demographic characteristics of households by dietary pattern in Bangladesh.

Characteristic	Rice and Low Diversity *n* (%)	Wheat and High Diversity *n* (%)	Pulses and Vegetables *n* (%)	Meat and Fish *n* (%)	Total *n* (%)
Total	263 (32.9)	262 (32.8)	259 (32.4)	16 (2.0)	800 (100.0)
Age group ^a^ (years)					
≤35	53 (20.2)	99 (37.8)	68 (26.3)	3 (18.8)	223 (27.9)
36–45	95 (36.1)	62 (23.7)	64 (24.7)	4 (25.0)	225 (28.1)
46–55	55 (20.9)	55 (21.0)	65 (25.1)	6 (37.5)	181 (22.6)
>55	60 (22.8)	46 (17.6)	62 (23.9)	3 (18.8)	171 (21.4)
Religion ^a^					
Muslim	247 (93.9)	223 (85.1)	230 (88.8)	11 (68.8)	711 (88.9)
Hindu/other	16 (6.1)	39 (14.9)	29 (11.2)	5 (31.3)	89 (11.1)
Educational level ^a^					
None	156 (59.3)	102 (38.9)	119 (45.9)	4 (25.0)	381 (47.6)
Primary school	96 (36.5)	123 (46.9)	113 (43.7)	7 (43.7)	339 (42.4)
Secondary/Tertiary	11 (4.2)	37 (14.1)	27 (10.4)	5 (31.3)	80 (10.0)
Household size					
1–4	89 (33.8)	159 (60.7)	119 (46.0)	7 (43.8)	374 (46.8)
5–7	136 (51.7)	86 (32.8)	116 (44.8)	8 (50.0)	346 (43.2)
>7	38 (14.6)	17 (6.5)	24 (9.3)	1 (6.3)	80 (10.0)
Farm production					
≤5	97 (36.9)	72 (27.5)	86 (33.2)	3 (18.7)	258 (32.2)
6–10	125 (47.5)	129 (49.2)	124 (47.9)	10 (62.5)	388 (48.5)
>10	41 (15.6)	61 (23.3)	49 (18.9)	3 (18.8)	154 (19.3)
Household wealth					
Poor	146 (55.5)	69 (26.3)	101 (39.0)	4 (25.0)	320 (40.0)
Medium	39 (14.8)	6.1 (23.3)	58 (22.4)	2 (12.5)	160 (20.0)
Rich	78 (29.7)	132 (50.4)	100 (38.6)	10 (62.5)	320 (40.0)
Region of residence					
Northern	83 (31.6)	50 (19.1)	59 (22.8)	8 (50.0)	200 (25.0)
Eastern	100 (38.0)	99 (37.8)	97 (37.4)	4 (25.0)	300 (37.5)
Central	39 (14.8)	50 (19.1)	49 (18.9)	2 (12.5)	140 (17.5)
Southern	41 (15.6)	63 (24.0)	54 (20.8)	2 (12.5)	160 (20.0)

^a^ Characteristic of household head.

**Table 2 nutrients-13-02049-t002:** Adequacy of macro-nutrient intake and fruit and vegetable intake by dietary pattern in Bangladesh.

	WHO Recommendation	Rice and Low Diversity (*n* = 263)	Wheat and High Diversity (*n* = 262)	Pulses and Vegetables (*n* = 259)	Meat and Fish (*n* = 16)	Total (*n* = 800)
Total energy (kcal/capita/day) mean (95% CI)		2812.86 (27310.8–2893.3)	3054.7 (2956.6–3152.8)	2918.7 (2839.1–2998.3)	3225.1 (2750.9–3699.3)	2933.5 (2883.5–2983.5)
Macro-nutrient (mean % of total energy)						
Carbohydrate	55–75	**78.9**	70.6	**76.1**	**75.2**	**75.2**
Fat	15–30	**9.1**	16.2	**11.4**	**12.5**	**12.2**
Protein	10–15	**9.0**	11.0	**9.7**	10.1	**9.9**
Fruit and vegetables (mean in grams)	400	**218.9**	**380.7**	**284.4**	**248.6**	**293.7**
Met WHO recommendation (%)						
Carbohydrate		100.0	97.7	100.0	100.0	99.2
Fat		3.0	48.1	8.5	25.0	20.0
Protein		10.0	37.0	29.0	37.5	25.4
Fruit and vegetables		3.4	36.6	15.4	12.5	18.4

Bolded values indicate inadequacy.

**Table 3 nutrients-13-02049-t003:** Predictors of dietary pattern in Bangladesh (mixed effects logistic regression).

Predictor	Rice and Low Diversity (*n* = 263)		Wheat and High Diversity (*n* = 262)		Pulses and Vegetables (*n* = 259)		Meat and Fish (*n* = 16)	
	AOR ^†^ (95% CI)	*p*-Value	AOR ^†^ (95% CI)	*p*-Value	AOR ^†^ (95% CI)	*p*-Value	UOR ^¶^ (95% CI)	*p*-Value
Household wealth		<0.001		<0.001		0.49		0.19
Poor	1		1		1		1	
Medium	0.46 (0.29–0.73)		1.94 (1.23–3.06)		1.21 (0.58–2.53)		1.00 (0.18–5.52)	
Rich	0.41 (0.27–0.61)		2.66 (1.77–4.02)		0.91 (0.54–1.51)		2.55 (0.79–8.21)	
Age group (years)		0.04		0.10		0.16		0.57
≤35	1		1		1		1	
36–45	1.75 (1.11–2.74)		0.63 (0.41–0.97)		0.87 (0.47–1.63)		1.32 (0.29–6.00)	
46–55	1.00 (0.61–1.64)		0.66 (0.42–1.04)		1.36 (0.51–3.66)		2.51 (0.62–10.20)	
>55	1.17 (0.71–1.92)		0.60 (0.37–0.98)		1.42 (0.47–4.31)		1.31 (0.26–6.57)	
Religion		0.02		0.25		0.84		0.03
Muslim	1		1		1		1	
Other	0.49 (0.27–0.91)		1.35 (0.81–2.25)		1.06 (0.58–1.93)		3.79 (1.28–11.17)	
Educational status		0.002		0.74		0.74		0.01
None	1		1		1		1	
Primary school	0.71 (0.50–1.01)		1.10 (0.77–1.57)		1.21 (0.63–2.31)		1.99 (0.58–6.80)	
Secondary/Tertiary	0.37 (0.18–0.77)		1.23 (0.71–2.15)		1.31 (0.48–3.63)		1.28 (1.65–23.95)	
Household size		<0.001		<0.001		0.80		0.93
1–4	1		1		1		1	
5–7	2.30 (1.59–3.32)		0.43 (0.30–0.62)		1.06 (0.70–1.62)		1.24 (0.45–3.46)	
>7	4.15 (2.28–7.58)		0.28 (0.15–0.54)		0.87 (0.41–1.83)		0.66 (0.08–5.47)	
Farm production		0.35		0.22		0.93		0.51
<5	1		1		1		1	
6–10	0.81 (0.56–1.18)		1.22 (0.83–1.80)		0.94 (0.61–1.44)		2.07 (0.56–7.60)	
>10	0.70 (0.42–1.17)		1.54 (0.94–2.52)		0.91 (0.50–1.64)		1.56 (0.31–7.81)	
Region of residence		<0.001		0.005		0.74		0.20
Northern	2.43 (1.46–4.03)		0.43 (0.26–0.70)		0.87 (0.45–1.69)		3.29 (0.69–15.72)	
Eastern	1.32 (0.81–2.17)		0.79 (0.50–1.25)		1.02 (0.61–1.70)		1.07 (0.19–5.89)	
Central	1.20 (0.68–2.12)		0.74 (0.44–1.25)		1.18 (0.56–2.48)		1.14 (0.16–8.24)	
Southern	1		1		1		1	

^†^ Adjusted for all other variables in the model (mixed effects model). ^¶^ Unadjusted odds ratio from univariate logistic regression.

**Table 4 nutrients-13-02049-t004:** Vulnerability scores of household dietary patterns.

Pattern	V_*1*_	V_*2*_	V_*3*_	Vulnerability Rank
Rice and low diversity	84.16	85.19	85.20	1 *
Wheat and high diversity	63.00	69.45	69.54	4
Pulses and vegetables	76.40	79.04	79.06	2
Meat and fish	75.50	78.81	78.81	3
Overall	74.55	77.92	77.95	

V*_i_* = proportion of total calories consumed from staple food accounted for by *i* most important staple crops. * Most vulnerable diet.

## Data Availability

The data used in this analysis have been made completely anonymized and available for free download by the International Food Policy Research Institute (IFPRI) through the following web address: https://doi.org/10.7910/DVN/27704, accessed on 10 June 2019. IFPRI obtained informed consent from all subjects involved in the study.

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
