# Peer review of "Characteristics of Distinct Dietary Patterns in Rural Bangladesh: Nutrient Adequacy and Vulnerability to Shocks"

_nutrients, 2021, doi:10.3390/nu13062049_

Round 1

Reviewer 1 Report

Thank you for your thorough reply to the comments, I have no further comments 

Reviewer 2 Report

The authors have well addressed my comments. I have no additional comments.

This manuscript is a resubmission of an earlier submission. The following is a list of the peer review reports and author responses from that submission.

Round 1

Reviewer 1 Report

Characteristics of distinct dietary patterns in rural Bangladesh: 2 nutrient adequacy and vulnerability to shocks

An interesting paper, very well-written and cohesive. I just have a few minor comments for consideration: 

  • Abstract, line 21: maybe better link the sentences by saying: "to inform this, we used a national dietary survey... "
  • Abstract, line 29: does this sentence apply to all the groups? Or just the pulses group?? Could you make this clearer “While energy intake was high, 29 fat and protein intake were sub-optimal.”
  • Abstract: You mention WHO recommendations – are these for adults or children? Or households? Its unclear from the abstract whether this is specific to adults or children.
  • Abstract: might be worth mentioning that “vulnerability to shock” was scored based on diversity of calorie sources. This wasn’t clear to me until I got to the methods section and I think it changes the interpretations of the findings slightly – its slightly more cyclical that vulnerability to shocks and dietary adequacy are related since diet diversity is relevant to both.
  • Introduction, line 51: I think these sentences could be improved and “would not be a welcome addition” seems a bit too informal for a scientific paper. Perhaps say “13% of the population 51 have insufficient habitual food consumption and 32% experiencing moderate or severe food insecurity [8]. In addition, Bangladesh is classified as being highly vulnerable to climate change [9] and has widespread subsistence agriculture [10] which adds further pressure to food production, supply and food prices.”
  • Methods, line 108: “Adequacy of energy intake from macronutrients was calculated following World 108 Health Organisation (WHO) guidelines on the appropriate daily range of proportions of 109 energy from macronutrients to total energy intake [19].” Again, do this recommendations not differ by age? Presumably this is for adults? Does sex matter? It would be good to have a little more detail on whether these are age and sex modified, or only applied to a certain group of household members i.e. adult women?
  • Methods, line 136: typo: ergy
  • Figure 1: the labels of the food groups on the X axis is very difficult to see, I suggest enlarging it as it took me a while to figure out what I was looking at
  • Results, line 227: you mention that some food groups met the calorie intakes recommended by WHO: “However, consumers of the Wheat & high diversity pattern did meet these requirements on average”. Given your mention of the double burden and high rates of obesity mentioned in the introduction, it might be worth spelling out here in the results whether any of these diets had excess intake putting them at risk of NCDs? You’ve told me that they aren’t deficient is some things but its not clear to me whether the richer food consumers are eating too much of anything.
  • Table 4: I think putting the 1, 2 and 3 in subscript in the table heading V1, V2 and V3 will make the interpretation of the footnote easier. Its not clear that Vi relates to V1, 2 and 3. Should be V1, V2
  • Discussion, line 332, “Even though these patterns were generally associated with high energy from 332 fats and proteins, the Wheat & high diversity pattern may be a healthier pattern than the 333 Meat & fish pattern because it had higher intakes of fruit and vegetables”. So does this mean that the groups representing further progression along the nutrition transition are not yet simailr to diets associated with high risk of NCDs? You mention the greater consumption of processed foods in some groups (line 342) – should you mention here an increased risk of NCDs? Also, Asian context are susceptible to NCDs are lower BMI’s than other global context, should this be mentioned in the discussion too? Your mention of NCDs in the final paragraph of the discussion implies that the wealthier diets are still at low risk of NCDs but still lacking in micronutrients – is that correct? Maybe good to spell out your position on NCD risk with some of the different groups because Im not quite clear at the moment.
  • Discussion, implications, line 416: “For example, there is large national production of poultry but consumption levels are lower especially for wealthy urban dwellers due to preference for ‘free living poultry’ which is expensive and not as widely produced” You say that poultry consumption is lower… lower than what? Also, the preference for free-living poultry has not been mentioned previously – this is not something you could have learned from this study – I suggest you reference it, and possibly mention it early in the discussion rather than springing a new concept into the implications section. Or maybe you can reword it so that it is a suggestion from the literature rather than a fact.
  • Discussion, implications: Im not confident about this comment, so feel free to ignore, but I wondered whether the pulses and vegetable diet might be an achievable short-term aim for low income households currently in the Rice and low diversity group? It is slight more nutrition and less vulnerable to shocks and also seems to be accessible to those across the income spectrum… ? Just an idea since not all household are going to be able to afford to move to the wheat and meat diet groups.
  • References: all the references beyond number 23 are missing from the reference list

Reviewer 2 Report

Thank you for the opportunity to review this manuscript. The authors identified main dietary patterns among rural residents in Bangladesh and potential determinants for these dietary patterns. The topic is very interesting; however, the authors may need to clarify some issues.

My major concerns:

  1. The authors used latent class analysis to identify dietary patterns. Can they report the percentage of each dietary pattern accounting for the total variance? These dietary patterns would be less valuable if they contribute to a minimal proportion of total variance.
  2. Latent class analysis is not a frequently used method to identify dietary pattern. Why did not they apply more frequently used factor analysis, cluster analysis, or reduced rank regression method to define dietary patterns?
  3. Amount of foods household members had consumed over the last 14 days preceding the survey were estimated by women. This estimation was not conducted day by day, such that I am not confident of the accuracy of this method. Is it validated? This needs to be discussed.
  4. Individual food intakes in four levels (zero, low, medium, and high) were used to identify dietary patterns. How were low, medium, and high levels defined? Why not use continuous variables in the analysis?
  5. Age, education, and religion of the household head were used to represent the whole household. Education and religion may be fine, but age of the household head may not be appropriate to represent the whole household level.

Minor concerns:

  1. The affiliation 2 (title page) was repeated.
  2. For Table 3, it is better to report the P-values for trend for dietary patterns associated with ordinary variables such as age, education, household size, and household wealth.
  3. Also for Table 3, the authors only reported the adjusted odds ratios but not unadjusted odds ratios. Frequency of participants with specific dietary patterns across subgroups of predictors including household health, age, and education may need to be presented.
  4. Were dietary intakes assessed in the same season among all participants? This is important as the availability of some foods such as fruits and vegetables are sensitive to the changes with season.